# Role of the Transcription Factor FOSL1 in Organ Development and Tumorigenesis

**DOI:** 10.3390/ijms23031521

**Published:** 2022-01-28

**Authors:** Vladimir V. Sobolev, Asiat Z. Khashukoeva, Olga E. Evina, Natalia A. Geppe, Svetlana N. Chebysheva, Irina M. Korsunskaya, Ekaterina Tchepourina, Alexandre Mezentsev

**Affiliations:** 1Center for Theoretical Problems in Physico-Chemical Pharmacology, Russian Academy of Sciences, 109029 Moscow, Russia; marykor@bk.ru (I.M.K.); tchepourina@mail.ru (E.T.); 2Federal State Autonomous Educational Institution of Higher Education, N.I. Pirogov Russian National Research Medical University of the Ministry of Health of the Russian Federation, 117997 Moscow, Russia; azk05@mail.ru; 3“JSC DK Medsi”, Medical and Diagnostics Center, 125284 Moscow, Russia; evinaolga@mail.ru; 4NF Filatov Clinical Institute of Children’s Health, I.M. Sechenov First MSMU, 119435 Moscow, Russia; geppe@mail.ru (N.A.G.); svetamma@gmail.com (S.N.C.)

**Keywords:** FOSL1/FRA1, tumorigenesis, cell differentiation, organ development, gene expression promoter, transcription factor

## Abstract

The transcription factor FOSL1 plays an important role in cell differentiation and tumorigenesis. Primarily, FOSL1 is crucial for the differentiation of several cell lineages, namely adipocytes, chondrocytes, and osteoblasts. In solid tumors, FOSL1 controls the progression of tumor cells through the epithelial–mesenchymal transformation. In this review, we summarize the available data on FOSL1 expression, stabilization, and degradation in the cell. We discuss how FOSL1 is integrated into the intracellular signaling mechanisms and provide a comprehensive analysis of FOSL1 influence on gene expression. We also analyze the pathological changes caused by altered *Fosl1* expression in genetically modified mice. In addition, we dedicated a separate section of the review to the role of FOSL1 in human cancer. Primarily, we focus on the *FOSL1* expression pattern in solid tumors, FOSL1 importance as a prognostic factor, and FOSL1 perspectives as a molecular target for anticancer therapy.

## 1. Introduction

FOSL1 protein is often considered a subunit of the transcriptional complex AP1. It plays a crucial role in cell differentiation, response to environmental stresses, and tumorigenesis. Including FOSL1, several proteins, primarily, members of JUN- and FOS-families contribute to AP1 by forming hetero- and homodimers. To the reference, JUN proteins can either homo- or heterodimerize, whereas, FOS proteins cannot homodimerize and have to heterodimerize, mainly, with JUN proteins.

Among AP1 subunits, there are three JUN (c-JUN, JUNB, and JUND) and five FOS (c-FOS, FOSB, ΔFOSB/FOSB2, FOSL1, and FOSL2) proteins. Moreover, the other proteins, such as ATFs (ATF1, ATF2, ATF3, and ATF4), NRFs (NRF1 and NRF2), and MAF may also contribute to AP1 (rev. in [1,2]). In addition, the published data suggest that some other transcription factors, such as NRL, JBP1, JBP2, and MYOD can compete with AP1 for AP1 binding sites. Thus, AP1 influence on the transcription is very complex and it depends on a delicate balance between many proteins, which are regulated by multiple signaling pathways. In turn, the transcription factor FOSL1 is a part of this multifaceted regulatory mechanism.

In the cell, FOS proteins contribute to the regulation of bone and lipid metabolism. Their abnormal expression in mice causes behavioral abnormalities, fibrosis, and cancer. This review aimed to summarize the available data on the role of the transcription factor FOSL1 in the regulation of gene expression, its contribution to the intracellular signaling mechanisms, and evaluate its clinical value as a molecular target and prognostic factor for anticancer therapy.

### 1.1. FOSL1 Binding to the DNA and Its Interactions with Other Transcription Factors

The AP1 heterodimers recognize two different palindromic sequences (rev. in [3]), namely TPA responsive elements (TRE) and cyclic AMP response elements (CRE). In addition, they can bind to variant sequences in the genome. The CRE sequence (A-T-G-A-C/G-G/C-T-C-A) is fully symmetrical while the TRE sequence (T-G-A-G/C-T-C-A), which is missing one of two central nucleotides, is pseudopalindromic. Both FOS-JUN and JUN-JUN dimers exhibit higher affinity to TRE and slightly lower affinity to CRE. The ability of AP1 to bind CRE suggests a crosstalk between AP1 and cAMP response element-binding protein (CREB), which also binds to CRE. Since different signaling pathways activate AP1 and CREB and rotating the transcription factor at the binding site allows the cell to tune up the transcription, this crosstalk has to regulate the gene expression. In contrast, the ATF-containing dimers preferentially bind to CRE when MAF-containing dimers bind to the extensions of TRE and CRE referred to as MARE I or MARE II, respectively [2].

Moreover, FOSL1 heterodimerizes with other transcription factors, such as the members of the bZIP family. Their hybrid heterodimers (e.g., FOSL1-USF2 [4,5]) can be transcriptionally inactive. Their possible role in the cell may include but not be limited to sequestering FOSL1 in the nucleus. Moreover, these dimers are either disabling the transcriptional activator complex or saving the interacting proteins from degradation in proteasomes. To date, there is no evidence that FOSL1 is capable of heterodimerizing with AP1 subunits other than JUN proteins (ATF, MAF, and FOS). On the other hand, FOSL1 can bind to several nuclear proteins that are not contributing to AP1. For instance, the complex of FOSL1, JUND, and SP1 acts as a repressor of ITGAV and ITGB3 genes binding to SP1 binding sites (5’-(G/T)-G-G-G-C-G-G-(G/A)-(G/A)-(C/T)-3’) in their promoters [6].

In summary, the ability of FOSL1 to dimerize with many partners can be important for three reasons. First, balancing the FOSL1 level in the cell allows the cell to tune up gene expression. For instance, a high FOSL1 level would facilitate the interaction of FOSL1 with some degenerate sequences that are similar but not identical to CRE and TRE and intensify the transcription. In contrast, a low FOSL1 level would restrict its interaction to canonical CRE and TRE elements. Second, optimizing the FOSL1 level makes it possible for different types of cells to react differently to the same stimulus. In turn, sequestering FOSL1 in transcriptionally inactive complexes would be important for the maintenance of FOSL1 concentration in the nucleus at a constant level as well as for destabilization of the transcriptional activator complex. Third, the interaction FOSL1 with different binding partners would allow the cell to control the transcription of more genes with fewer numbers of specialized proteins.

### 1.2. Tissue-Specific Expression of FOSL1 Gene

In the cell, FOSL1 typically resides in the nucleus. However, in certain conditions, such as oxidative stress, FOSL1 can be detected in the cytoplasm [7,8]. Some authors report the cytoplasmic staining for FOSL1 in cancer cells [9]. In this case, the cytoplasmic FOSL1, presumably, is needed to activate the biosynthesis of phospholipids [8], which are required for the migrating cancer cells to support their rapid membrane biogenesis. In turn, neutralizing FOSL1 with specific antibodies slows the biosynthesis of phospholipids down. In embryonic development, *FOSL1* expression is higher than in postnatal life [10,11]. Compared to the other adult tissues, a relatively high *FOSL1* expression can be observed in the brain [12], the heart and skeletal muscles [13,14], pancreas, and bones [15]. In the lung and the skin, *FOSL1* expression increases at the sites of inflammation [16,17]. Moreover, *FOSL1* upregulation often occurs during tumorigenesis.

As a proto-oncogene, *FOSL1* can be induced by many inflammatory cytokines and mitogens mainly through the RAS signaling pathway. Notably, many authors report a constitutive activation of RAS in cancer cells. In vitro, *FOSL1* is expressed by leukocytes [18], fibroblasts [19], adipocytes [20], and many other cells. Cellular stress, such as bleeding, ultraviolet radiation, and genotoxic agents, sequentially induces FOS genes, including *FOSL1* (rev. in [2]).

## 2. Stabilization and Degradation of FOSL1 in the Cell

### 2.1. Role of Mitogen-Activated Protein Kinases in Stabilization of FOSL1

FOSL1 participates in several signaling cascades (Figure 1) that are activated by various growth factors, such as FGF2, VEGF, and insulin (rev. in [21]). On the other hand, only three protein kinases, namely ERK2, RSK2, and ERK5, interact with FOSL1 directly. Neither JNK [22] nor MAPK p38 [23,24] seems to have any role in the phosphorylation of FOSL1. In the human FOSL1, ERK2 phosphorylates Ser252 and Ser265, ERK5 phosphorylates Thr230 and RSK phosphorylates Ser265 [22,25] (Figure 2). In all three cases, phosphorylation stabilizes the FOSL1 molecule, saving it from the degradation in proteasomes [19]. Moreover, threonine phosphorylation increases FOSL1 biological activity [22]. For instance, changing Thr231 (the murine homolog of human Thr230) for Asp produces a dominant negative form of Fosl1. This mutant Fosl1 fails to induce Fosl1 target genes, such as Mmp1 [23].

### 2.2. FOSL1 Phosphorylation by PKCθ

Despite FOSL1 stabilization in tumor cells that express constitutively activated RAS and RAF mostly depends on ERK-induced phosphorylation [26], high levels of phosphorylated FOSL1 can be observed even if the cells do not express constitutively active mutant forms of RAS or RAF [27]. For instance, phosphorylation and stabilization of FOSL1 in the estrogen receptor (ER)—negative breast cancer cells occurs due to crosstalk between RAS and PKCθ and involves the protein kinase SPAK1 [28]. In this respect, either knocking PKCθ down or overexpressing PKCθ dominant negative isoform results in a dramatic reduction of FOSL1 level, whereas PKCθ overexpression leads to FOSL1 phosphorylation and accumulation in the cells. In turn, the repression of both ERKs and SPAK1 causes a decline of the FOSL1 level by 60%, whereas the individual inhibition of either ERKs or SPAK1 produces only a slight effect on FOSL1 accumulation. This observation suggests that ERKs and SPAK1 contribute independently to FOSL1 stabilization. Since FOSL1 does not possess the consensus motif [S/G/V]RFx[V/I]xx[V/I/T/S]xx [29], which is characteristic for SPAK1 substrates, it is more likely that SPAK1 acts via another kinase rather than directly phosphorylates FOSL1. Although the details of PKCθ-dependent FOSL1 phosphorylation still need to be revealed, it is already known that FOSL1 becomes phosphorylated on Ser-265, Thr-223, and Thr-230 [21].

### 2.3. Degradation of FOSL1 in Proteasomes

Although FOSL1 is a protein with predominantly nuclear localization, sometimes FOSL1 can be found in the cytoplasm. The cytoplasmic FOSL1 may undergo polyubiquitination. Then, it degrades in proteasomes [30]. The degradation of FOSL1 in proteasomes is mediated by a subunit of the 19S proteasome regulatory particle referred to as TBP1 (PRS6A/Rpt5/S6). To the reference, TBP1 participates in the degradation of both ubiquitinated and non-ubiquitinated proteins [31]. Moreover, it does not interact in the same way with other FOS proteins (e.g., c-FOS). The latter suggests that FOSL1 has a unique mechanism of degradation in proteasomes. Specifically, TBP1 binds to the middle of the FOSL1 molecule (residues 162–231 in human FOSL1 [32]).

The degradation of nuclear FOSL1 may occur regardless of its ubiquitination status [33] because the C-terminal area of the FOSL1 molecule (residues 231–271) is unstable. For its specific role in FOSL1 degradation, this part of the FOSL1 molecule is known as c-DEST/DESTAB [34]. The c-DEST area (Figure 2) does not contain any element of the secondary protein structure. Like other proteins containing c-DEST, FOSL1 is recognized as a misfolded protein and is subjected to disposal. In turn, the export of FOSL1 to the cytoplasm may occur due to an interaction of c-DEST with specific nuclear chaperones that recognize the dephosphorylated C-terminus of FOSL1, unfold the whole molecule, and transport unfolded FOSL1 to the cytoplasm [30].

A necessity in the ubiquitin-independent mechanism can be explained in two ways. First, non-ubiquitinated FOSL1 would be much easier to unfold and transport to the cytoplasm. Second, the cell expresses FOSL1 at certain stages of the cell cycle. Specifically, it peaks in G_0_/G_1_ and coincides with the activation of ERKs. Thereafter, FOSL1 level declines sharply. Thus, the cell needs a specific mechanism to perform this rapid FOSL1 decline.

## 3. Role of FOSL1 in Regulation of Gene Expression

### 3.1. Assessment of FOSL1 Binding Sites in the Genome

In the genome of cultured cells with a constitutively activated Raf/Mek/Erk pathway, the total number of potential Fosl1 binding sites may reach 11,000–13,000 [35,36]. Expectedly, a waste majority of these sites is associated with potential regulatory areas of the genome, such as promoters, introns, and distant enhancers, whereas only ~1% of the proposed Fosl1 binding sites are located in the gene-coding areas [35]. According to Diesch et al., who studied human BE colon carcinoma cells, ~72% percent of the proposed FOSL1-regulated genes are 5-fold enriched for FOSL1 binding sites within 5 kb of their transcription start site (TSS) [36]. In turn, the ratio of FOSL1 binding sites that match the canonical TRE and CRE sequences is nearly 58% [35]. This finding is not very surprising for three reasons. First, we may not yet know all FOSL1 binding partners [4,6]. Second, some non-canonical AP1 sites that FOSL1 binds may become available only at specific physiological conditions, such as hypoxia [37]. Third, Fosl1 binding may require specific cluster compositions [38,39,40] or polymorphism [41]. In this respect, the likelihood and nature of particular binding sites have to be examined on an individual basis to reveal ones that can be crucial for the transcription.

Although the transcriptome analysis provides important information about molecular interactions between FOSL1 and its target genes, it does not allow us to say whether FOSL1 directly interacts with a particular gene promoter or whether the effect of FOSL1 on gene expression is mediated indirectly, i.e., via FOSL1 target genes. Moreover, direct and indirect biological effects on the same gene often enhance each other. For instance, FOSL1-dependent induction of cyclin D1 (CCND1) occurs due to the induction of the transcription factor *DMP1* that FOSL1 controls.

According to microarray and next-generation sequencing data, either *FOSL1* overexpression or silencing influences hundreds of genes. For instance, *Fosl1* overexpression in mouse mammary epithelial cells EpH4 by the factor 1.5 induces 400 genes and suppresses 250 genes [42]. In turn, knocking *FOSL1* down by ~75% in the human triple-negative breast cancer cells BT549 results in the upregulation of 263 genes and downregulation of 156 genes [43]. Similar results are obtained in HUVEC [6] and BE colon carcinoma cells [36].

### 3.2. Clustering FOSL1 with Other Transcription Factors and Participation in the Transcription Factors Network

Like many other transcription factors, FOSL1 either induces or represses the gene expression, i.e., FOSL1 influence on a particular gene primarily depends on the FOSL1 level in the cells and FOSL1 binding partners. It also depends on the transcriptional microenvironment of the FOSL1 binding site [44]. Binding AP1 to the DNA integrates FOSL1 into a cluster of several transcription factors [2]. The transcription factors contributing to the cluster have DNA binding sites in proximity of each other. Respectively, when these transcription factors bind to the DNA, they are supposed to fit a relatively narrow space like pieces of a puzzle directly interacting with each other. Their direct interaction improves the structural stability of the cluster. It also makes it possible for FOSL1 containing AP1 dimers to interact with non-canonical TRE and CRE sites.

As a specific transcriptional microenvironment, the cluster is dynamic and cooperative in its nature. This means that many participating transcription factors, including FOSL1, are replaceable by other molecules on a competitive basis. Consequently, the participating transcription factors act as a block. They also represent an integral part of the transcriptional activator complex, which recruits chromatin-modifying proteins (e.g., p300 [45], MSK1, EZH1, BRG1/SNF2L4 [46], and KDM4A/JMJD2A [47]) and interacts with RNA polymerase II [48].

For instance, the activation of a highly conserved *IL6* promoter depends on the cooperation of several transcription factors, namely AP1, CREB, C/EBP, and NFκB (Figure 3A). In this respect, the contribution of each transcription factor to IL6 expression may sufficiently vary between different types of cells because the cooperation between AP1 and the other transcription factors relies on a specific set of cofactors, which is unique for any cell kind [38,46]. In IM9 cells that overexpress IL6, deletion of either NFκB or CREB sites does not change the promoter activity. Contrarily, disabling AP1 or C/EBP binding sites inhibits the transcription [38]. In human intestinal epithelial Caco2 cells, AP1, CREB, and C/EBP cooperate on the control of the gene. In contrast, NFκB does not influence the gene expression [39]. At the same time, all four sites are required for a constitutive *IL6* expression in PC3 and DU145 prostate cancer cells [40].

In some cases, AP1 binding to the promoter occurs only in certain physiological conditions. For example, some researchers considered MMP2 (Figure 3B) as PMA’s unresponsive gene (rev. in [49]). In normal physiological conditions, the expression of MMP2 is under the control of other transcription factors. These transcription factors bind to the promoter in three distinct regulatory areas. The first area, known as RE1, contains binding sites of several transcription factors, namely Yb1, AP2, and p53. These transcription factors produce a synergistic effect on the transcription [50,51]. Contrary, the cells cultured in hypoxia respond to PMA by a strong induction of *MMP2* [37]. Moreover, the PMA-dependent induction of *MMP2* can be even enhanced by co-stimulation of the cells with angiotensin II, endothelin I or IL1β.

Since the preliminary experimental data indicated the involvement of AP1 in the regulation of *MMP2*, the investigators performed a reporter assay that pointed them to a non-canonical AP1 site located in the proximity of RE1. According to their data, this site responds to stimulation with PMA. Moreover, their experiments demonstrate cooperation between AP1 and the other transcription factors that bind to RE1 [37]. In addition, another group reports that a replacement of Fosl1-Junb by Fosb-Junb heterodimers at the *Mmp2* promoter can be relevant to skeletal muscle atrophy [52]. They show that the binding of Fosl1 to the Mmp2 promoter increases with the progression of the disease and results in the induction of Mmp2.

Remarkably, even minor changes in the promoter sequence, such as SNP, may influence the promoter response to AP1. For instance, the polymorphism in the *MGP* promoter known as rs1800802 (T_-138_-C) introduces an additional non-canonical AP1 binding site (T-G-A-C-T_-138_-G-T) between two other regulatory elements, namely E-box and C/EBPβ (Figure 3C). This site binds FOSL1, FOSL2, c-JUN, and JUNB containing AP1 heterodimers [41]. Moreover, partial overlapping of a retinoic acid response element (RAR) with the C/EBPβ binding site suggests that RAR may act as a potential repressor of this improvised regulatory element similarly to how it does in the IVL promoter (rev. in [53]).

Notably, the following observations discovered that both variations, i.e., T_-138_ and C_-138_, are widely represented in the human population [41]. Taking into account an inhibition of vascular mineralization by MGP, we can speculate that the polymorphism rs1800802 could be beneficial for the carriers of the T-138 genotype due to a higher MGP activity in their blood. Respectively, a proposed delay in the progression of the disease can be associated with excessive calcification of blood vessels and internal organs. Surprisingly, the studies aimed to confirm the proposed beneficial role of the T_-138_ genotype in selected cardiovascular [54,55], kidney [56], and bone [57] conditions do not reveal any health risk for C_-138_ carriers. The statistical analysis of the experimental data performed by their authors suggests that healthy carriers of the T-138 allele, indeed, exhibit higher MGP levels in the blood [41,54,55]. Moreover, the differences in calcification between these two groups of carriers are insignificant [54].

The *FOSL1* promoter is inducible by various growth factors and cytokines (rev. in. [21]). In this respect, the *FOSL1* promoter (Figure 3D) reminds a hub that contains binding sites of many transcription factors, such as AP1, ELK1, SRF, and ATF/CrREB absorbing signals of multiple signaling pathways. Moreover, the first intron of the FOSL1 gene contains another regulatory area with a canonical AP1 binding site (rev. in [21]). The existence of additional AP1 binding sites makes possible the autoregulation of FOSL1. This site is involved in the epithelial-mesenchymal transition (EMT) of cancer cells requiring a stable and intensive FOSL1 expression (see. below). Expectedly, FOSL1 induction also requires extensive chromatin remodeling, such as changes in the histone pattern and subsequent recruitment of chromatin-modifying proteins [58].

The transcription factor Fosl1 is a part of a network of transcription factors. This network also includes the transcription factors Hmga2, Otx1, Klf6, Gfi1, Junb, and Rela [59]. These transcription factors regulate each other at both transcriptional and posttranscriptional levels. Most of the named transcription factors, such as Junb and Fosl1, can be controlled by the network via induction or suppression of their encoding genes. Some others, like Rela, are mainly regulated by the network via the signaling pathways. The latter means that the transcription factors involved in the network change the expression of growth factors and cytokines, which activate the underlying signaling mechanisms (e.g., Mekk and Pi3k). In turn, the activated signaling mechanisms either activate or inhibit Rela.

Although the entire network is not extensively analyzed, the investigators suggest that this network is decentralized [35]. In other words, none of the tested transcription factors, including Fosl1, can be a master key that connects separate hubs and prevents the network from falling apart. Notably, about 80% of genes involved in the network are controlled in a combinatorial manner by several transcription factors. Moreover, the analyzed transcription factors produce a unidirectional effect on the waste majority of differentially expressed target genes. Thus, a randomly chosen differentially expressed gene will be either induced or repressed by all these transcription factors.

In turn, a computer analysis of the network divided the transcription factors that we mentioned above into two groups with different levels of competence. Three transcription factors, namely Rela, Gfi1, and Otx1, appear upstream of four others (Fosl1, Junb, Hmga2, and Klf6). According to the investigators, members of the first group contribute to the regulation of each other. Moreover, they control members of the second group. In addition, they exhibit stronger influences on cell growth, migration, and proliferation. Moreover, silencing Otx1, Gfi1, or Rela causes the cell cycle arrest in the G_1_-phase. On the contrary, members of the second group, i.e., Fosl1, Junb, Hmga2, or Klf6, did not significantly affect the expression of Otx1, Gfi1, and Rela. They also produce milder biological effects on cell growth, migration, and proliferation. In addition, their silencing does not cause a cell cycle arrest.

### 3.3. The Role of FOSL1 in Epithelial-Mesenchymal Transition

Overexpression of *FOSL1* in immortalized cells causes rapid changes in cell morphology [42]. These changes are a part of EMT (Figure 4). In growing tumors, EMT coincides with budding. During the budding, tumor cells detach from the invasive front of the tumor and invade into the basement membrane [60]. Then, the cells enter the bloodstream and establish a population of circulating tumor cells. In turn, this precedes the development of metastases [61]. After disseminating across the body, the circulating tumor cells undergo a mesenchymal-epithelial transition (MET), i.e., they regain epithelial phenotype and colonize the invaded tissues [62].

In agreement with these findings, *Fosl1* overexpressing immortalized epithelial cells suppress epithelial biomarkers, such as cadherin E (Cdh1), and induce endothelial biomarkers, such as vimentin (Vim), Ascdh2, fibronectin (Fn1), Cdh3, S100a4, Spp1. They also upregulate the genes encoding integrins α5 and β1 (*Itgba5* and *Itgb1*, respectively) [42]. Moreover, these cells overproduce the enzymes that are involved in the degradation of the extracellular matrix (ECM) and proteins of intermediate filaments (e.g., Mmp1a, Mmp2, Mmp9, and Mmp14, uPA/Plau, and uPAR/Plaur) [42]. The Fosl1 overexpressing epithelial cells also downregulate the genes encoding the proteins of tight and adherens junctions, namely Crb3, Cldn23, Tjp1, Ocln, Jup, etc. In turn, their downregulation impairs the maintenance of cell polarity and paracellular transport. In addition, Fosl1 overexpression leads to downregulation of the cytokeratins Krt7 and Krt8. It also causes downregulation of the integrins β2, β5, and β6 (Itgb2, Itgb5, and Itgb6, respectively). Thus, the intracellular contacts between *Fosl1* overexpressing cells, such as tight and adherens junctions, get loose [36]. The cells develop numerous protrusions and tend to migrate. After transplantation to the mice, these cells efficiently colonize the lung and transform into rapidly growing and highly vascularized tumors.

The sharp change of cell morphology during FOSL1-induced EMT would be impossible if the response did not involve a widespread network of transcription factors and was not highly coordinated [35,59]. The gene ontology analysis suggests that both *FOSL1* overexpression [42] and *FOSL1* silencing [35,36,43] influence the genes involved in two major categories, namely EMT and cell adhesion. These genes encode a diverse array of proteins that contribute to signal transduction, transcription, and motility as well as cytoskeletal and extracellular matrix remodeling.

Unlike *Fosl1* overexpressing cells, *Fosl1*-silenced cells suppress the genes associated with the mesenchymal phenotype, including the EMT promoting transcription factors *Snail*, *Slug*/*Snai2*, *Zeb1*, and *Zeb2*, and induce the genes expressed by the epithelial cells (e.g., *Cdh1*) [6]. Moreover, knocking *Fosl1* down in poorly differentiated epithelial tumor cells restores their epithelial morphology [42,43]. The cells change their shape and flatten out. They also suspend their invasive behavior, lose tumorigenicity [36,43] and restore tight junctions [36].

The assembly of their adherens junctions and cytoskeleton in FOSL1-deficient cells is severely impaired. For instance, *FOSL1*-silenced HUVEC cells accumulate ανβ3 integrin dimers that cluster on the cell surface [6] and develop significantly more focal adhesion sites than the parental cell line. In turn, clustering ανβ3 induces relocation and concentration of non-receptor cytoplasmic proteins on focal adhesion sites that control the cytoskeletal organization and downstream signaling. Respectively, the rearrangement of these proteins alters the spatial organization of focal adhesion kinase (FAK/PTK2) and paxillin (PXN). It affects the recruitment of cytoskeleton and signaling proteins and influences a coordinated transmission of the downstream signals. Unlike their parental cells, where actin filaments form parallel bundles, the actin filaments of transformed cells look highly disoriented. Changes caused by silencing *FOSL1* slow down FAK turnover and inhibit cell migration. The FOSL1-deficient cells lose their ability to self-organization. They fail to develop a pseudocapillary network. In turn, re-establishing *FOSL1* expression to a certain extent resolves the mentioned cytoskeletal abnormalities and normalizes the level of ανβ3. It also restores cell motility and their ability to form pseudocapillaries.

The data obtained on Hs578T and EpH4 cells [42,43] suggest that the biological effects of FOSL1 on cell differentiation as well as the development of adherents and tight junctions are mediated, in part, by the transcription factor ZEB2, which is one of the FOSL1 target genes. Silencing *ZEB2* in *FOSL1*-deficient cells restores the epithelial phenotype inducing *CDH1*, *CTNNB1*, and *CTNND1* [42]. Moreover, silencing *ZEB2* decreases cell motility and tumorigenicity.

In many malignant cells that overexpress *FOSL1*, upregulation of the genes associated with the mesenchymal phenotype coincides in time with activation of TGFβ-signaling. In this respect, there is an opinion that the TGFβ-signaling pathway mediates FOSL1-induced EMT in the cultured cells. This hypothesis is supported by the following data. First, the TGFB1 promoter discovers several transcriptionally active AP1 binding sites. Second, *FOSL1* overexpression in cultured cells results in upregulation of TGF receptors agonists *TGFB1* [42] and *TGFB2* [36]. It also causes downregulation of TGF receptors antagonists *BMP4* and *BMP7* [36]. Third, upregulation of several pro-mesenchymal FOSL1 target genes (e.g., *AXL* [36], *VIM* [36,42], and *TGFB1* [36]) in *FOSL1* overexpressing cells follows the activation of the TGFβ signaling. However, the other data suggest that the TGFβ signaling only partially mediates FOSL1 biological effects. Primarily, the inhibition of TGF receptors in *FOSL1* overexpressing cells does not affect cell morphology. However, it causes a moderate increase in the expression of some epithelial markers [42]. It also does not affect the expression of pro-epithelial FOSL1 target genes *CDH1* and *CLDN7*. In this respect, the exact role of the TGFβ signaling in EMT has yet to be clarified. In the first turn, the biological effects that FOSL1 produces due to the activation of the TGFβ signaling still needs to be sorted out from the others.

Thus, the experimental data suggest that FOSL1 induces the genes required for EMT and the following migration and invasion of the cancer cells. The published results prove the importance of FOSL1 for coupling the oncogenic RAF/RAS/ERK/RSK and other pro-tumorigenic signaling mechanisms, such as the TGFβ pathway with EMT. In this respect, FOSL1 induces the genes associated with the mesenchymal (invasive) phenotype. Moreover, FOSL1 overexpression causes downregulation of the genes expressed in the epithelial cells. Primarily, it downregulates the genes involved in the formation of intracellular contacts, such as tight and adherens junctions.

### 3.4. Changes in Phenotypes of Fosl1-Deficient and -Overexpressing Mice

The studies of genetically modified mice provide valuable information on how Fosl1 contributes to cell differentiation and organ development (Figure 5). The conventional genetic deletion of *Fosl1* results in embryonic lethality [10] because Fosl1 is crucial for the conversion of embryonic stem cells to trophoblasts [63] and the development of the placenta [10]. The fra-1^Δ/Δ^ mice lacking *Fosl1* in embryonic tissues are viable and fertile [11]. At birth, these mice are indistinguishable from their Fosl1-sufficient littermates. This observation suggests that *Fosl1* is dispensable in embryonic tissues. In several weeks, the animals develop osteopenia [11], a bone condition caused by the abnormal functioning of osteoblasts. The bones of *Fosl1*-deficient mice elongate, and the bone mass reduces. The analysis of changes in gene expression suggests that Fosl1 controls the bone-forming activity of osteoblasts. The late differentiation marker of osteogenesis, Osteocalcin (Bglap), is downregulated. In contrast, the expression of other bone differentiation markers, namely Runx2, Pdlim3/Alp Ibsp, Rankl, and Opg, is changed insignificantly [11]. Thus, the observed bone phenotype of conditional *Fosl1* knockout mice is similar to low-turnover osteoporosis in humans.The inflammatory response developed by Fosl1-deficient mice after their treatment with lipopolysaccharides (LPS) is milder than in wild-type animals [64]. In the experimental model of acute lung injury, the *naïve Fosl1*-deficient animals develop fewer lung injuries. They also demonstrate a higher survival rate compared to control [64,65]. The expression and secretion levels of proinflammatory cytokines (Mip2, Mip2α/*Ccl3*, Il1β, and Tnf) are reduced [64]. Moreover, their immune infiltrates contain fewer neutrophils and macrophages than these cytokines attract. In addition, the concomitant deletion of *Fosl1* in *Fosl1*^F/F^: *Kras*^G12D^ mice overexpressing the *Kras* oncogene reduces tumorigenesis in the lung. It also increases their survivability. These results suggest the downregulation of *Fosl1* as a potential treatment option for anticancer therapy [66].

In contrast to *Fosl1* knockout mice, *Fosl1* transgenic mice (*H2-Fosl1-LTR*) develop osteosclerosis, a bone disorder, which causes a progressive increase in bone mass of the entire skeleton [67]. Although Fosl1 transgenic mice are viable and fertile, they do not live longer than nine months due to severe splenomegaly [67]. Moreover, the transgenic mice spontaneously develop liver fibrosis [68], bronchoalveolar tumors [67], impaired adipogenesis, and lipodystrophy [69]. In addition, *Fosl1* transgenic mice also demonstrate a delay in the healing of bone fractures, due to impaired chondrogenesis [70].

Comparative analysis of gene expression in *Fosl1* transgenic and wild-type mice suggests that Fosl1 promotes the differentiation of progenitor cells into osteoblasts rather than adipocytes without challenging the mesenchymal cell commitment [69]. The transgenic mice exhibit a low level of Cebpa, which is the key factor of late adipogenesis. At the same time, the expression of genes that regulate differentiation of progenitor cells to osteoblasts (*Ctnnb1* and *Runx2*) and early stages of adipocyte differentiation (*Cebpb* and *Cebpg*), as well as the markers of adipocytes progenitor cells (*Cd24a*, *Cd34*, *Ly6a*, and *Itgb1*) remains unchanged. Moreover, the cultured osteoblasts exhibit high levels of the osteogenic growth factors, such as Bglap, matrix Gla protein (Mgp), and the major bone collagen Col1a2 [11].

According to Hasenfuss et al. [71], lipodystrophy in *Fosl1* transgenic mice can be caused by downregulation of *Pparg2*, which is a key regulator of fatty acid storage and glucose metabolism. The investigators report that Pparg2 expression depends on the composition of AP1 heterodimers that bind to the Pparg2 promoter. They found that enrichment of AP1 binding sites in c-Fos and Jund accelerates Ppparγ signaling. Contrarily, enrichment of these sites in Fosl1 and Fosl2 inhibits the signal transduction through the Pparγ signaling pathway. Respectively, overexpression of Fosl1 would delay the accumulation of lipids via suppression Pparγ signaling pathway.

Moreover, the transgenic adipocytes express less of the fatty acid transporter Cd36, which regulates fatty acids uptake and lipoprotein lipase (Lpl) that hydrolyzes triglycerides in lipoproteins. Presumably, the decreased expression of *Cd36* and *Lpl* explains hypertriglyceridemia in the blood serum of transgenic animals. The expression level of Glut4, a biomarker of mature white adipose tissue, is also decreased [69]. The genes encoding adipokines adiponectin (*Adipoq*) and resistin (*Retn*) are slightly downregulated. In contrast, the gene of leptin (*Lep*) is significantly suppressed due to the immaturity of the adipocytes.

The mice that overexpress Fosl1 develop spontaneous liver fibrosis, the disease associated with ductular proliferation and sustained inflammation [68]. The induction of proinflammatory chemokines (Cxcl5, Ccl1, Ccl5, Ccl8, and Ccl20) causes infiltration of the liver by immune cells. One of these chemokines, Ccl20 is a chemoattractant of lymphocytes. Ccl1, Ccl5, and Ccl8 attract T- and B-cells. Ccl5 and Ccl8 attract eosinophil granulocytes. Cxcl5 stimulates the chemotaxis of neutrophils and enhances angiogenesis. In addition, Fosl1 induces some receptors of the mentioned chemokines (Cxcr1, Ccr2, and Ccr4).

The inflammation and bile duct proliferation in the liver cause a loss of bile ducts. Respectively, the development of the disease leads to an activation of profibrotic genes (*Col1a1*, *Col1a2*, and *Col3a1*). The profibrogenic cytokines, namely transforming growth factor β1 (*Tgfβ1*) and *Pdgfd*, which are Fosl1 target genes, are strongly induced at the early stages of the disease. In contrast, Pdgfb is induced at the later stages of the disease. The expression of Pdgfa and Pdgfc remains unchanged. The transgenic mice also exhibit a higher alkaline phosphatase (Alp) activity, which is common in many biliary and liver disorders.

In conclusion, we would like to acknowledge the importance of Fosl1 for the differentiation of adipocytes, chondrocytes, and osteoblasts. Controlling, primarily, late stages of cell differentiation, Fosl1 is indispensable for the normal development of the bones, the liver, and the immune system. In this respect, both *Fosl1* deficiency and overexpression cause severe abnormalities in mice.

## 4. Role of FOSL1 in Tumorigenesis

### 4.1. FOSL1 Expression Pattern in Solid Tumors

Since metastatic tumors usually originate from epithelial cells, blocking EMT in tumor cells with a epithelial phenotype is crucial to prevent metastasis. Since EMT reverses cell differentiation and induces stem-like traits, the cells experiencing EMT lose their apicobasal polarity and escape the anchorage-dependent surveillance systems. These cells detach and migrate through the basal membrane, invading and settling down in healthy tissues. Moreover, the EMT-transformed cells lower their proliferation rate and develop drug resistance [72,73]. Then, the survived cancer cells produce metastases and transform into new tumors [74,75].

According to immunohistochemical data, FOSL1 can be detected in nearly 100% thyroid [76], 87% esophageal carcinomas [77], 67% bladder tumors [78] and 92% breast carcinoma samples [79]. In addition, one of the most aggressive forms of cancer, malignant glioma, is also characterized by high AP1 transcriptional activity and, particularly, by higher levels of FOSL1 [80].

Despite intense staining for FOSL1 in tumor cells, the healthy epithelia either remain FOSL1-negative or express FOSL1 at low levels [36,79]. Typically, the marginal area of the tumor, which is adjacent to the site of inflammation, exhibits higher staining. Then, the staining declines toward the tumor center [36,81]. Moreover, higher FOSL1 levels may indicate the presence of metastases since specimens taken from patients with metastases in the liver [36] and the lymph nodes [82] exhibit more intense staining for FOSL1 than the tumor samples taken from the patients that do not have metastases.

The time course of FOSL1 expression may vary in different kinds of solid tumors. For instance, esophageal and colon tumors are characterized by high levels of FOSL1 at early stages, whereas thyroid and breast tumors continue to accumulate FOSL1 even at late stages of the disease. The numerous clinical studies suggest several important correlations and specific characteristics of the tumorigenesis related to FOSL1.

In breast cancer, a higher FOSL1 expression is often observed in the ER-negative tumors, i.e., the most aggressive and highly malignant breast carcinomas, such as HER2-positive [83] and triple-negative breast carcinomas (TNBC) [84]. Moreover, the EMT-transformed breast cancer cells often accumulate FOSL1, regardless of their ER status, preparing for EMT and tumor dissemination. In malignant gliomas [85], thyroid [86], and some other tumors, higher FOSL1 levels are associated with the development of a malignant phenotype. The latter is accompanied by altering the shape of tumor cells and impairing their anchorage. For instance, FOSL1 accumulation in non-tumorigenic glioma cell lines in vitro confers tumorigenicity [87]. On the contrary, diminishing the FOSL1 level by 80% restores cell morphology, impairs cell motility, and normalizes the metabolic processes [88].

In summary, FOSL1 overexpression is characteristic for the most aggressive forms of cancer, such as TNBC and malignant glioma. It may indicate metastases in the liver and lymph nodes. In solid tumors, the FOSL1 staining pattern is not uniform. The invasive front of the tumor discovers more intense immunohistochemical staining for FOSL1. In contrast, the tumor core shows significantly weaker staining. Higher FOSL1 expression is associated with EMT, which precedes tumor dissemination, and plays a crucial role in the survival of the cancer cells.

### 4.2. FOSL1 as a Prognostic Tool

The new prognostic biomarkers are in high demand because reliable prognostic biomarkers for some tumors are not yet identified. Moreover, there is a rising interest in new prognostic tools that could improve the method as well as optimize the treatment strategy for a particular patient. Obviously, FOSL1 can be used as a prognostic tool for several reasons. First, strong staining for FOSL1 is often observed in solid tumors of a higher grade. Second, tumors with higher FOSL1 levels are frequently associated with aggressive forms of cancer. Third, FOSL1 overexpression can be linked to resistance to chemotherapy. In this respect, monitoring the FOSL1 level could help to define the tumor grade [79] and choose the most efficient therapeutic strategy for individuals with advanced cancer [89]. Notably, FOSL1 is already included in the tri-molecular signature for glioblastoma, together with IL13Rα2 and EPHA2 [85].

In breast cancer, different subtypes of breast tumors, including estrogen receptor α (ERα)-positive tumors, discover high expression levels of FOSL1. The results of immunohistochemistry analysis demonstrate a positive correlation of FOSL1 level with the tumor grade. For instance, the neoplastic cells of the advanced TNBC and atypical malignant tumors usually exhibit moderate-to-strong nuclear staining for FOSL1 [90]. At the same time, the differences in FOSL1 levels and copy numbers between HER2-enriched and TNBC are not significant [91,92,93]. The highest FOSL1 expression occurs in metastatic lesions of lymph nodes where it is about 50% higher than non-metastatic tumors. Moreover, higher FOSL1 levels are observed in tumor-associated macrophages [89,94].

On the contrary, FOSL1 is detectable only in one-third of the primary low-grade breast tumors. In this respect, only ~10% of the preserved ductal cells of these tumors are positive for FOSL1 [9,91]. In premalignant non-nodular breast lesions, FOSL1-specific staining is always evident around ductal structures in the proximity of the basal membrane [91], whereas healthy ductal cells are usually FOSL1-negative. In addition, non-proliferative lesions, including papillary lesions, apocrine metaplasia, and columnar alterations, are also FOSL1-negative.

Notably, the microarray data obtained on a large cohort of breast cancer patients suggest that *FOSL1* mRNA levels have a significant inverse correlation with the patients’ survival [42,66]. Moreover, the prognosis for the patients with higher *FOSL1* expression can be even worse if a high *FOSL1* expression coincides with elevated expression of *JUN* in the same tumor [43,95]. Among the genes encoding FOS proteins, this inverse correlation is unique for *FOSL1* [43], whereas upregulation of either *FOS* or *FOSB* is associated with a slightly better outcome for the patients [90].

Although FOSL1 expression occurs in epithelial and mesenchymal colorectal tumors, the mesenchymal cells discover more intense staining for FOSL1 than the epithelial cancer cells [36]. Similar to the breast tumors, the FOSL1 level positively correlates with tumor grade and suggests poorer survival of the patients with colorectal cancer. In addition, the incorporation of FOSL1 and FOSL1-target genes associated with EMT (e.g., ZEB1) into the disease-predicting signature makes the predictions of the disease recurrence risk and the risks of some adverse effects more accurate.

In esophageal squamous cell carcinomas, FOSL1 expression is typically higher in patients with advanced cancer [77]. FOSL1 expression correlates with the depth of the tumor, the presence of metastases in the lymph nodes, stage, and aggressive behavior [77,96]. Moreover, the experimental data suggest FOSL1 as an independent prognostic factor for the patients’ survival time. In this respect, a higher accumulation of FOSL1 in the primary tumor and metastatic lymph nodes suggests a poorer survival prognosis to the patient. On the contrary, patients with FOSL1-negative tumors and metastatic lymph nodes may survive more than five years after the surgical resection under the average survival rate of less than 25%.

In bladder cancer, more intense staining for FOSL1 tends to be more prominent in EMT-transformed invasive cells [78]. The intensive nuclear staining for FOSL1 is more often in advanced tumors. In contrast, non-malignant bladder urothelium discovers no nuclear staining for FOSL1. Moreover, the intensity of FOSL1 immunostaining negatively correlates with the survival rate of the patients.

### 4.3. FOSL1 as a Molecular Target for Anticancer Therapy

Developing new therapeutical approaches to target FOSL1 has at least two advantages. First, both tumor cells and tumor-associated immune cells often express FOSL1. In contrast, the surrounding epithelial cells either remain FOSL1-negative or exhibit low FOSL1 levels. Second, FOSL1 expression is higher in EMT-transformed tumor cells that can potentially enter the bloodstream and invade healthy tissues. Accordingly, depleting FOSL1 is supposed to initiate MET in these cells and make them susceptible to chemotherapy.

Unfortunately, FOSL1 is not likely to be amenable to pharmacologic inhibition by small molecules [97]. However, several options that involve unconventional mechanisms can be used to suppress and destabilize FOSL1 in cancer cells. Primarily, the extensive studies of FOSL1 phosphorylation suggest that FOSL1 can be destabilized and depleted by inhibition of the upstream kinases, primarily RAS and ERK (rev. in [89]). The inhibition can be ineffective for several reasons. First, the RAS/RAF/ERK/RSK pathway initiates multiple underlying metabolic and signaling cascades (rev. in [98]), i.e., kinases that contribute to FOSL1 stabilization and induction (except BRAF) can phosphorylate as many as hundreds substrates. In this respect, targeting a single protein will affect multiple physiological responses directly unrelated to FOSL1 producing severe adverse effects if the inhibitor is used for a long time. Second, the RAS/RAF/ERK/RSK pathway is indispensable for healthy cells. Although most of the kinase inhibitors are tolerable by healthy cells (rev. in [98,99]), targeting cancer cells in experimental animal models may produce a surprising systemic response in multiple organs and tissues including the liver, the brain, and the immune system (rev. in [98]). Third, cancer cells may easily bypass the targeted kinase due to a rapid accumulation of new mutations, even if a combination of several inhibitors is used (rev. in [98,100]). In this case, they can reactivate the affected pathway by establishing crosstalk with another signaling mechanism (rev. in [98]). Moreover, a continuous accumulation of mutations by a targeted kinase may make the inhibitor ineffective (rev. in [101]). Fourth, the cancer cells may amplify the expression of inhibited kinases and improve their sensitivity to other cytokines and growth factors [102].

Alternatively, *FOSL1* mRNA can be destroyed with one of the endogenous mi-RNAs prior to it reaching the ribosomes. Similarly to protein-coding genes, the genes that encode mi-RNAs can be regulated via regular signaling mechanisms by the same transcription factors. For instance, the equilibrium between epithelial and mesenchymal states is governed by self-enforcing double-negative feedback loops that are composed of the essential for EMT transcription factors and several mi-RNA. Particularly, the repression of ZEB1 and ZEB2 by ectopic expression of either miR-221 or -222 restores CDH1 expression [103,104,105], whereas both ZEB1 and ZEB2 repress the expression of these miRNAs [106,107]. Similarly, both miR-34b/c (two mi-RNAs that have the same transcript) and miR-34a are directly repressed by SNAIL1 [108,109] and ZEB1 and vice versa [110]. The abundance of these miRNAs is decreased with the inhibition of MEK (mitogen-activated or extracellular signal-regulated protein kinase), placing miR-221/222 downstream of the RAS pathway. Moreover, the expression of miR-34a, miR-221, and miR-222 in metastatic breast cancer tissues negatively correlates with FOSL1 expression, although there is no correlation between FOSL1 and miR-34b/c levels [111].

Notably, *FOSL1* mRNA carries the binding sites of the following miRNAs-miR-34a-c [111,112], -221, -222 [21], miR-149 [113], miR-138 [114]. Respectively, in the cells that overexpress the mentioned mi-RNA, FOSL1 mRNA will likely be degraded. On the other hand, each mi-RNA has multiple target genes, i.e., FOSL1 is not going to be the only gene that mi-RNA destroys. For instance, miR34a and miR34b/c target *CCNE2*, *AXL*, *LMNB2*, *CDK4/6*, *NOTCH1*, *E2F3*, *MYCN*, *SIRT1*, *MET*, *CCND1*, *BCL2*, *ZEB1*, etc., interfering with the normal flow of cell proliferation and causing cell cycle arrest in the G_1_ phase and promoting cell apoptosis and senescence. In this respect, overexpression of these mi-RNAs will likely promote the development of adverse effects (rev. in [115]).

In this respect, using artificially designed sh-RNAs (small hairpin RNAs) will help achieve a more specific FOSL1-silencing. Moreover, prefabricated viral particles deprived of their virulent factors will deliver specific sh-RNA to the cancer cells and produce a stable biological effect, i.e., silencing FOSL1 will be permanent. In this case, the sh-RNA encoding sequence will be integrated into the cell genome and constitutively transcribed. To date, the efficiency and specificity of viral transfection are already proven in the experiments on cultured cancer cells (rev. in [116]). In addition, a modification of viral particles will increase the specificity of their interaction with the desired subpopulations of tumor cells. For instance, the modified virions may carry ligands that would specifically recognize receptors on the surface of cancer cells.

In summary, FOSL1 expression is detectable in the brain, heart, skeletal muscles, pancreas, and bones. It is common in solid epithelial tumors despite healthy epithelia may express FOSL1 at a low level or remain FOSL1 negative. A higher expression of FOSL1 occurs in undifferentiated budding cells. In contrast, the differentiated tumor cells that reside in the core of the tumor discover significantly weaker staining. The metastatic tumors express more FOSL1 compared to benign tumors. Respectively, the specific staining for FOSL1 is more intense in tumors of higher grades. The mesenchymal tumors may express FOSL1 at a significantly higher level than the tissue of their origin. This type of tumor also shows more intensive staining for FOSL1 than respective epithelial tumors. In addition, a higher expression of FOSL1 occurs in tumor-associated macrophages. The latter suggests that the role of FOSL1 in tumorigenesis extends beyond EMT. As already proposed, FOSL1 may contribute to remodeling the extracellular matrix [117] and modulates the expression of chemokines [118].

## 5. Conclusions

The regulation of gene expression by the transcription factor FOSL1 has several layers of complexity. FOSL1 dimerizes with different binding partners and interacts with various binding sites of the DNA. Its transcriptional activity can be tuned by neighboring transcription factors on the cluster. It can be stabilized by several protein kinases and, unlike other FOS proteins, FOSL1 has a unique mechanism of degradation in proteasomes. Although FOSL1 is dispensable in embryonic tissues, the conventional knockout of FOSL1 is lethal because it plays a crucial role in the development of the placenta. FOSL1 is also important for the differentiation of the progenitor adipocytes, chondrocytes, and osteoblasts. It participates in the formation and growth of bones. Disabling *Fosl1* in mice leads to osteoporosis. Contrarily, its overexpression results in splenomegaly, liver fibrosis, and bronchoalveolar cancer.

Speaking of cancer, we have to mention the decisive role of FOSL1 in EMT. Increasing the FOSL1 level in epithelial cells induces the genes associated with the mesenchymal phenotype. Contrarily, suppression of FOSL1 restores their morphological traits, impairs motility, and normalizes the metabolic processes. Moreover, a low FOSL1 level reduces the intensity of the inflammatory response. Since FOSL1 is associated with most malignant forms of cancer and is often considered a poor prognostic factor for the outcome, it makes FOSL1 an attractive target for anticancer therapy. Presumably, targeting FOSL1 could establish a strong barrier for the EMT tumor cells and prevent the appearance of metastases. In addition, the therapeutic reduction of the FOSL1 level has to be less risky, since FOSL1 seems to be dispensable in adults.

## Figures and Tables

**Figure 1 ijms-23-01521-f001:**
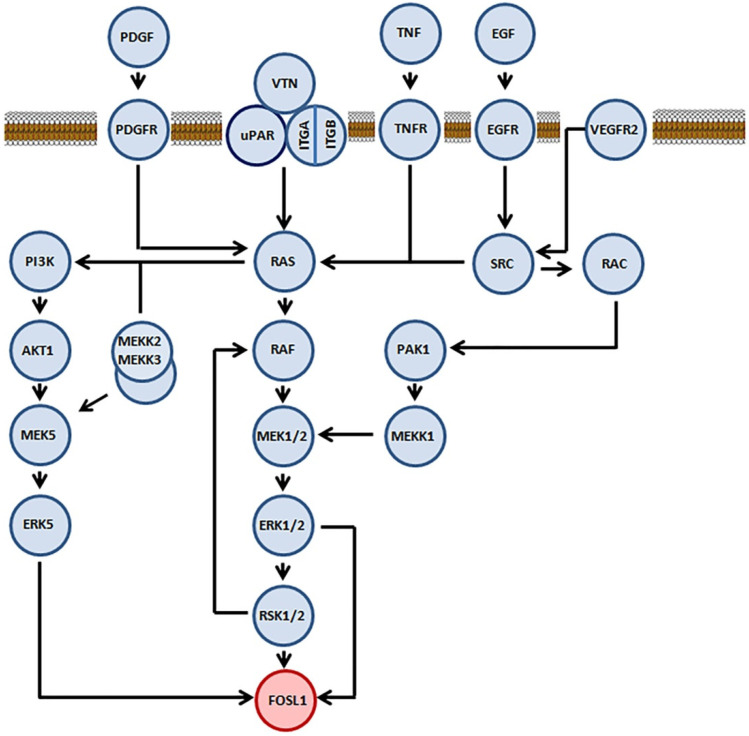
Participation of the transcription factor FOSL1 in the intracellular signaling pathways. Proteins involved into the signaling cascade that FOSL1 directly participates are marked in light blue. FOSL1 is marked in pink. The lipid bilayer shown in the upper part of the figure is the cellular membrane.

**Figure 2 ijms-23-01521-f002:**
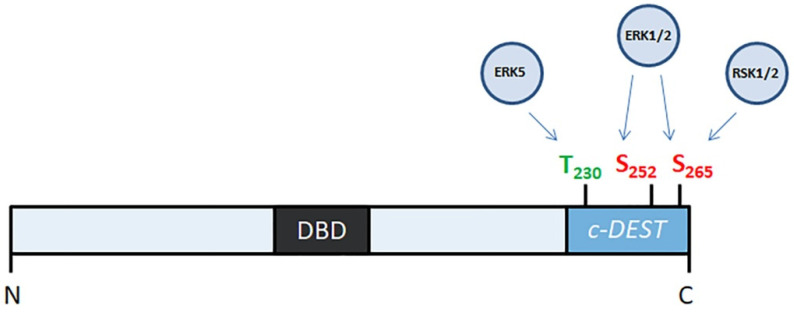
Phosphorylation FOSL1 by protein kinases. N- and C- ends of FOSL1 molecule are indicated as “N” and “C”. DBD—DNA-binding domain; c-*DEST*—C-terminal unstructurized destabilizing area.

**Figure 3 ijms-23-01521-f003:**
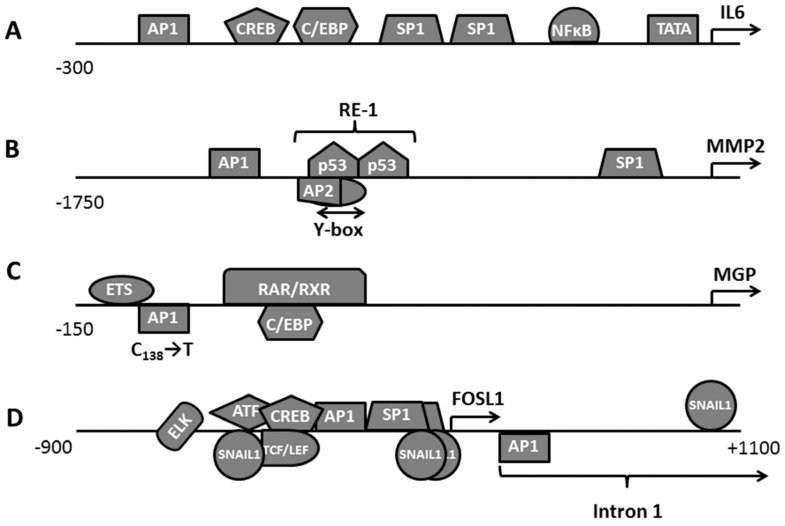
Regulation of the selected gene promoters by the transcriptional complex AP1: (**A**)—*IL6*; (**B**)—*MMP2*; (**C**)—*MGP*; (**D**)—*FOSL1*.

**Figure 4 ijms-23-01521-f004:**
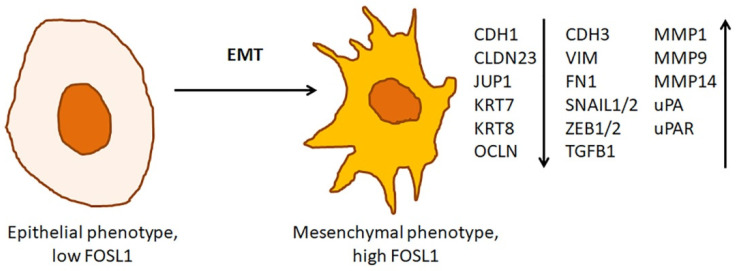
Role FOSL1 in epithelial-mesenchymal transition. Overexpression of FOSL1 in immortalized cells changes the cell morphology. The epithelial cells lose polarization and cell-to-cell contacts. They also suppress the expression of epithelial biomarkers. Instead, they induce the expression of endothelial biomarkers and proteolytic enzymes capable of degrading the components of the extracellular matrix. The arrows ↑ and ↓ indicate whether a particular gene is up- or downregulated, respectively, in the cells with the mesenchymal phenotype.

**Figure 5 ijms-23-01521-f005:**
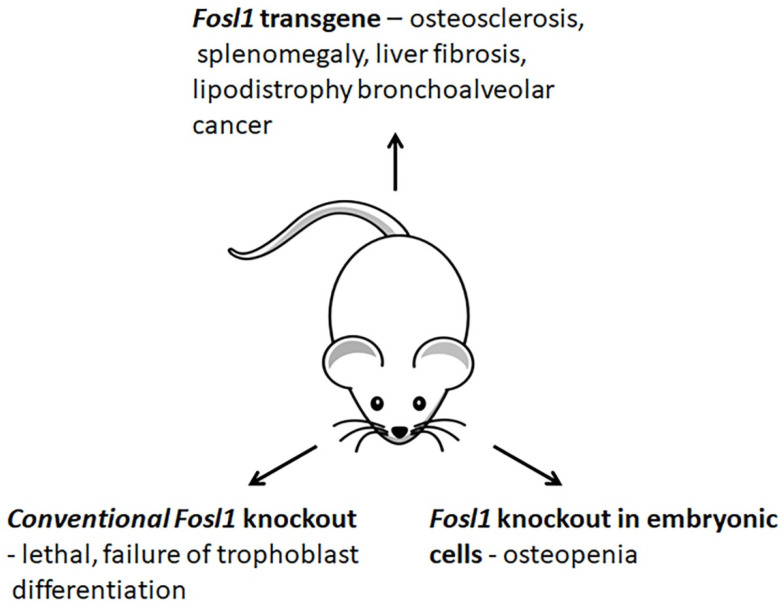
Changes in phenotypes of Fosl1-deficient and -overexpressing mice.

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
