# Peer review of "Role of the Transcription Factor FOSL1 in Organ Development and Tumorigenesis"

_ijms, 2022, doi:10.3390/ijms23031521_

Round 1

Reviewer 1 Report

The article reports many aspects of FOSL1/FRA1 activity and regulation  in cells, mouse models and in human tumors, ranging from binding to DNA and other transcription factors to degradation. Despite the effort of the authors in summarizing them, in my opinion the review aims to address too many aspects of FOSL1 function and regulation, and results long, fragmentary, in some parts too general and in others too detailed in reporting findings and conclusions from specific articles. Therefore, it is not focused, and, in particular, it is not specifically focused on the oncogenic role of FOSL1/FRA1 in tumor, that, instead results not fully addressed. Other reviews have been recently published on specific aspects of the role of FRA1 in tumorigenesis (see Jiang et al., 2020; Talotta et al., 2020) that in my opinion better present the topic.  

Moreover, some concepts are misleading and/or confused and not adequately supported by references (i.e. lines 55-63, lines 76-79).

  • Lines 55-63: As reported in the review of Bejjani and coworkers (2019) “Depending on their compositions, AP-1 dimers bind to different types of palindromic sequences. Thus, Fos:Jun and Jun:Jun dimers preferentially bind to DNA motifs referred to as 12-O-tetradecanoylphorbol-13-acetate (TPA)-responsive element (TRE; also called AP-1 motif) and, with however slightly lower affinity, cAMP-responsive element (CRE). The consensus sequences for TRE/AP-1 and CRE are 5′-TGA(C/G)TCA and 5′- TGACGTCA motifs, respectively…..On their side, ATF-containing dimers preferentially bind to cAMP-responsive element (CRE) whereas MAF-containing dimers bind either to MARE I or MARE II motifs that are extensions of TRE and CRE motifs, respectively”. I would suggest to clarify this part (the TRE is the proper AP1 binding site) and add the relative references.
  • Lines 75-81: I would move the lines 79-81 and link them to line 74, reporting first the possible bindings and then summarizing their impact. Moreover, please clarify the significance of FOSL1 binding to several partners (i.e. points 1 and 2 are not clear to me, whereas in point 3 I would highlight the potential to regulate other genes outside the AP1 pathway). I would also report in this list the possibility to sequester or be sequestered in inactive transcriptional complexes and its significance.

Regarding the structure, the order, the title, and subdivision of the paragraphs should be improved. There are also two paragraphs with the same title (2.1 and 2.2) that could, in fact, be merged. Conclusions refer only to the last part of the review.

I would recommend a careful consideration on the aims of the review and the message to give to readers, and then  an extensive general revision, taking into account for example the following points:

  • eliminate and/or extensively shorten some aspects (i.e. stabilization and degradation; mouse models),
  • fuse some paragraphs. For example, the expression pattern during embryo development and adult life might be fused to expression in tumors. A discussion on the type of tissue in which Fra1 is normally expressed and the tumors in which it is overexpressed might be also added. Also paragraph 3 on regulation of gene expression might be moved directly after Introduction and Table 1, in which the different transcription factors and binding sites are described.  
  • highlight the functional role of FRA1 in EMT (maybe paragraph 3.2 might be devoted to this and entitled accordingly) as well as in other aspects of tumorigenesis, such as cell proliferation and inhibition of apoptosis.
  • improve conclusions that should reflect the entire article content

Figures and Tables need to be improved. 

Figure 1 is complex and not so explicative (arrows, symbols and abbreviations are not described), and a way to differ between TFs, extracellular signals, receptors and intracellular signaling molecules would be useful. Moreover, Figure 1 seems a bit out of the context, inserted in the paragraph “Role of mitogen-activated protein kinases in stabilization of FOSL1”. The figure is not explained at all, nor in the text neither in the capture. As it is, the Figure does not have special significance. Maybe other Figure(s) representing more the text should be added. Table 1 seems more a list than a table and should be revised.

Finally, the article needs careful language editing and proofreading.

Below only some points:

Line 19: “osteoblasts” or “osteocytes”?

Line20: I would suggest to use “transition” instead of “transformation”

Lines 20-21: please clarify the sentence” gaining stem cell properties….”

Line 22: I would suggest to substitute “known facts” with “present in the literature” or “evidence”

Line 52: “FOSL1 (italic) expression”

In many other cases author use “FOSL1 expression”. Expression refers to the gene, if authors want to refer to protein, better use “protein levels “, otherwise they should put FOSL1 in italics.

Line 86: I would suggest to delete the first sentence, too general.

Line 87: I would suggest to substitute “for the life time” with “ post-natal life”.

Line 296: “waste majority”

Line 327 “EMT-transformed cells”

Line 417 “Fosl1 is dispensable for embryonic development”. It is dispensable in embryonic tissues, whereas it is required in extraembryonic ones for embryo development.

Names of the different factors should be specified the first time that abbreviations are used or in alternative specification of the abbreviations might be  inserted at the end of the article.

Author Response

Dear Editor,
On behalf of my colleagues, I would like to thank both reviewers for the time that they devoted to our manuscript, their comments, and suggestions.

Looking through the opinion written by reviewer #1, we discovered 10 comments that required our response.
Comment 1.1 (lines 53-63 in non-revised manuscript) was regarding the affinity of TRE and CRE.
Answer 1.1: We cited Bejjani's paper (Ref. 3). We also mentioned the difference in affinities of CRE and TRE (lines 58-59 in the revised manuscript). We also named MARE1 and MARE2 as the binding sites of MAF-containing AP-1 dimers (lines 64-65) and CRE as well as the binding site of ATF-containing AP-1 dimers (lines 63-64).

In Comment 1.2 (lines 73-81 in non-revised manuscript), the reviewer wanted us to clarify the significance of FOSL1 binding partners and suggested and suggested us to make several rearrangements in the text.
Answer 1.2: I complied with the reviewer's request and corrected the text accordingly (lines 80-91).

Comment 1.3 was regarding the structure, the order, the title, and subdivision of sections 2.1 and 2.2 (lines 112 and 132, respectively).
Answer 1.3 We renamed section 2.2 and split the former section 3.2 into two parts, the new sections 3.1. and 3.3, respectively. In the new section 3.1 (lines 178-207) we discuss the assessments of FOSL1 binding sites in the genome. In turn, the new section 3.3 describes the role of FOSL1 in epithelial-mesenchymal transition (lines 322-418).

Comment 1.4:was regarding the aims of our paper.
Answer 1.4: We clarified the aims of our paper and shortened the sections of our manuscript dedicated to stabilization and degradation of FOSL1 (Sections 2.1 and 2.3).
Moreover, we would like to explain why we would avoid further cuts to these sections. Reason 1 is that we want to highlight the role of phosphorylation of serine and threonine residues in the stabilization of FOSL1. At the same time, it seems to have no role in the activation of FOSL1. We wanted the readers who are not familiar with this topic to understand it and do mix stabilization and activation. For this reason, we provided details on the phosphorylation of FOSL1. Reason 2 is that FOSL1 has a unique mechanism of degradation. This mechanism is unique among FOS proteins. Respectively, we want the readers to understand why the cell does not accumulate a significant amount of FOSL1 and how it gets rid of this protein in the cell cycle.

Comment 1.5 was regarding fusing and rearranging the text in section 1.2 Tissue-specific expression of the FOSL1 gene.
Answer 1.5: we corrected the text accordingly (lines 93-100). The updated section 1.2 contains 2 paragraphs and it is only 18 lines long.

Comment 1.6 was regarding the former section 3.2 The role of FOSL1 in epithelial-mesenchymal transition.
Answer 1.6: we complied with the reviewer's request and split the former section 3.2 into two parts: the new sections 3.1 Assessment of FOSL1 binding sites in the genome and 3.3 The role of FOSL1 in epithelial-mesenchymal transition.

In comment 1.7 the reviewer wanted us to clarify the Conclusions section.
Answer 1.7: we complied with the reviewer's request and wrote the Conclusions over (lines 676 -697).

In comment 1.8 the reviewer wanted us to put Figure 1 in the context of the manuscript.
Answer 1.8: We simplified Figure 1, and modified it according to the reviewer's request. In the new Figure 1, we preserved only the pathways that are directly involved in the phosphorylation of FOSL1.

Comment 1.9 was regarding the content of Table 1.
We removed Table 1 and made several minor adjustments to the text.

Comment 1.10: the reviewer suggested us to perform proofreading and editing of the manuscript.
We complied with the reviewer's request and corrected the text. In this regard, we would like to make several minor remarks.
a) "osteoblasts" or "osteocytes" (former line 19). To our knowledge, osteoblast lineage cells include both osteoblasts and osteocytes. Respectively, if we refer the readers to osteoblast lineage, we may mean "osteocytes".
b) about the replacement of “known facts” with “present in the literature” or “evidence” (former line 22). We replaced “known facts” with “available data”.
c) We placed the list of abbreviations after "Conclusions" (lines 698-734 in the revised manuscript).

Reviewer 2 Report

This manuscript reviews the role of FOSL1, a transcription factor in the AP1 family, in the development and tumorigenesis. The authors provided a comprehensive review of the complex function of FOSL1. Here are the major and minor comments to improve this review article.

Major

  • lack of effective illustrations: the current review should significantly be improved by adding a few key illustrations that depict the role of FOSL1 in organ development as well as tumorigenesis.
  • genome-wide analysis: the role of FOSL1 as a transcription factor can be presented by genome-wide transcriptome analysis. There is a description but no presentation of critical data is given.
  • functions beyond a transcription factor: FOSL1 is located not only in the nucleus but also in the cytoplasm. FOSL1 may have multiple functions besides acting as a transcription factor. It is recommended to describe other potential functions.
  • EMT and transcription: The question is whether the role of FOSL1 in EMT is mainly through transcription. The authors are recommended to provide data on this topic.
  • binding site: The affinity to any specific base varies among specific base pairs in the binding site. The authors should mention this probabilistic nature of binding sites.

Minor

  • Table 1: This table is not comprehensive and its significance is questionable.
  • knockout vs. transgenic: The difference is mentioned but without knowing the specific transgenic model it is difficult to understand the difference.
  • five FOS: it is recommended to add the description that compares the role of 5 FOS transcription factors.

Author Response

Dear Editor,

On behalf of my colleagues, I would like to thank both reviewers for the time that they devoted to our manuscript, their comments, and suggestions.

Looking through the opinion written by reviewer #2, we noticed 10 comments that required our answer.

Comment 2.1. The reviewer wanted us to provide several key illustrations that depict the role of FOSL1 in organ development and tumorigenesis.

Answer 2.1. We complied with the reviewer's request and included four additional figures to the text of the revised version of the manuscript. We also simplified Figure 1.

Comment 2.2. The reviewer wanted us to present a genome-wide transcriptome analysis.

Answer 2.2. In response to the reviewer's comment, we would like to acknowledge that this kind of analysis would be nearly impossible to perform because gene expression is tissue-specific and any tissue of the body has its own transcriptome. Instead, we presented available data on the tissue-specific expression of FOSL1 (lines 92-110 in the revised version of the manuscript) and the role of FOSL1 in the regulation of gene expression (lines 178-207 in the revised version of the manuscript).

Comment 2.3. The reviewer wanted us to comment on the role of FOSL1 in the cytoplasm.

Answer 2.3. To satisfy the reviewer's request, we commented on the role of FOSL1 in the cytoplasm. For your convenience, please see lines 95-100 in the revised version of the manuscript.

Comment 2.4. The reviewer wanted us to provide the data on the role of FOSL1 in epithelial-mesenchymal transition (EMT).

Answer 2.4. In response to the reviewer's comment, we would like to acknowledge that the revised version of the manuscript contains a two-page section where we describe the role of FOSL1 in EMT. For your convenience, please, see lines 322-418 and new Figure 4 in the revised version of the manuscript.

Comment 2.5. The reviewer wanted us to explain how the composition of FOSL1 binding sites might affect the affinity.

Answer 2.5. To satisfy the reviewer's request we provided the available data on the affinity of FOSL1 JUN-JUN and JUN-FOS dimers to CRE and TRE and included the references to the relevant papers. For your convenience, please see lines 58-59 in the revised version of the manuscript and Ref. 3 for the details.

Comment 2.6. The reviewer questioned the significance of Table 1

Answer 2.6. Respectively, we removed the former Table 1 to satisfy with the reviewer's request.

Comment 2.7. The reviewer wanted us to indicate the names of animal strains that the other researchers used in their experiments with knockout and transgenic animals.

Answer 2.7. We satisfied the reviewer's request and added the names of animal strains to the text of section 3.4, which describes changes in phenotypes of Fosl1-deficient and –overexpressing mice (lines 420-502).

Comment 2.8. The reviewer suggested us to comment on the role of other FOS proteins in the cell.

Answer 2.8. To address the reviewer's concern, we provided a brief description of the role of FOS proteins in the cell (lines 46-47 in the revised version of the manuscript).

Round 2

Reviewer 1 Report

The authors have carefully revised and improved the manuscript, including the Figures. They have addressed almost all my comments,  just missing to discuss the relationship among the type of tissue in which Fosl1/Fra1 is normally expressed in the embryo or adult life and the tumors in which it is overexpressed. It would be interesting to have some comments.

In addition, very few things:

  • Add in Figure 1 caption something about the cell membrane that has been inserted in the Figure;
  • In Figure 5 I would suggest using “FosL1 KO in embryonic cells”,
  • Please make a final spell check (i.e. line 86 “interaction of FOSL1; Figure 5 “osteosclerosis”; ).

Author Response

Dear Editor,
On behalf of my colleagues, I would like to thank both reviewers. Their comments and suggestions helped us significantly improve our manuscript. To comply with the reviewer's #1 suggestions, we modified the capture of Figure 1, updated Figure 5, and added a brief comment regarding the relationship of the tissues expressing FOSL1 and tumorigenesis (lines 657-669). We also proofread the manuscript to improve its readability and correct misspellings.

Reviewer 2 Report

The authors responded to the comments to the original manuscript satisfactorily. The manuscript is now acceptable after a minor spell check.

Author Response

(The authors gave the same response as above.)
